# CheXseg: Combining Expert Annotations with DNN-generated Saliency Maps for X-ray Segmentation

**Soham Gadgil**[*]                                              SGADGIL@STANFORD.EDU

**Mark Endo**[*]                                              MARKENDO@STANFORD.EDU

**Emily Wen**[*]                                               EMILYWEN@STANFORD.EDU

**Andrew Y. Ng**                                                  ANG@CS.STANFORD.EDU

**Pranav Rajpurkar**                                         PRANAVSR@CS.STANFORD.EDU

*Department of Computer Science, Stanford University*

## Abstract

Medical image segmentation models are typically supervised by expert annotations at the pixel-level, which can be expensive to acquire. In this work, we propose a method that combines the high quality of pixel-level expert annotations with the scale of coarse DNN-generated saliency maps for training multi-label semantic segmentation models. We demonstrate the application of our semi-supervised method, which we call CheXseg, on multi-label chest X-ray interpretation. We find that CheXseg improves upon the performance (mIoU) of fully-supervised methods that use only pixel-level expert annotations by 9.7% and weakly-supervised methods that use only DNN-generated saliency maps by 73.1%. Our best method is able to match radiologist agreement on three out of ten pathologies and reduces the overall performance gap by 57.2% as compared to weakly-supervised methods.

**Keywords:** Semi-Supervised Segmentation, Saliency Maps, Localization Performance

## 1. Introduction

The "black box" nature of neural networks represents a barrier to physicians' trust and model adoption in the clinical setting (Kelly et al., 2019). Saliency maps are a popular set of explanation methods that highlight regions of the image that are important for disease classification, but they have been shown to be untrustworthy for medical image interpretation (Eitel and Ritter, 2019; Crosby et al., 2020; Young et al., 2019; Arun et al., 2020). Segmentation models can produce more accurate pixel-level maps, but their training is typically limited by the time-consuming process of collecting expert annotations. The combination of saliency maps generated from widely available classification models and a limited amount of expert annotations for training medical image segmentation models may be able to provide higher quality segmentations at a lower cost, but this approach remains relatively unexplored.

In this work, we develop *CheXseg*, a semi-supervised method for multi-pathology segmentation that leverages both the pixel-level expert annotations and the saliency maps generated by image classification models. First, we find that CheXseg achieves a mean IoU of 0.270, outperforming both fully-supervised (mIoU of 0.246) and weakly-supervised (mIoU of 0.156) methods. Second, we find that initializing the segmentation encoder with weights

---

[*] Contributed equally

learned from supervised classification of the same task leads to higher performance than using a self-supervised MoCo initialization (He et al., 2020) or ImageNet initialization (Deng et al., 2009). Third, CheXseg reduces the overall gap to radiologist localization performance (mIoU) by 57.2% compared to solely using DNN-generated saliency maps. We expect this method to be broadly useful for medical image segmentation, where classification models are widely available and expert annotations are expensive.

## 2. Related Work

### 2.1. Weakly-Supervised Semantic Segmentation

In this work, we focus on an approach in which classification models trained with image-level labels are used to create pixel-level pseudo-labels (Yao and Gong, 2020; Ciga and Martel, 2019; Ouyang et al., 2019). These pseudo-labels can then be utilized to train a segmentation model. Our paper is closely related to Viniavskyi et al. (2020), which proposes a deep CNN-based approach that generates pseudo-labels by applying an Inter-pixel Relation Network (IRNet) (Ahn et al., 2019) to improve Grad-CAM++ (Chattopadhay et al., 2018) generated activation maps. This approach is evaluated on the SIIM-ACR Pneumothorax dataset. In our work, we generate pseudo-labels with IRNet and extend the approach to the semi-supervised setting for a larger set of pathologies.

Several other methods also propagate class activation from areas of high confidence to similar adjacent regions (Kolesnikov and Lampert, 2016; Huang et al., 2018; Ahn and Kwak, 2018). We choose to build upon Viniavskyi et al. (2020) rather than these approaches because it has the best performance on the PASCAL VOC 2012 (Everingham et al., 2010) validation set (mIoU of 0.646) and was shown to perform well on a medical imaging task.

### 2.2. Semi- and Fully-Supervised Semantic Segmentation

Semi-supervised methods use a combination of expert pixel-level annotations and pseudo-labels to train semantic segmentation models. Some weakly-supervised methods have been extended to semi-supervised methods through the replacement of pseudo-labels. The weakly-supervised SGAN model (Yao and Gong, 2020) was adapted to a semi-supervised setting by replacing a subset of the saliency maps with the corresponding manually annotated ground truth labels. In this work, we use a similar idea of utilizing radiologist annotated labels in addition to saliency maps to train the segmentation model, extending Viniavskyi et al. (2020)'s weakly-supervised model.

Though weakly- and semi-supervised methods can perform well, fully-supervised methods are still considered the upper-bound (Chan et al., 2020). Many fully-supervised semantic segmentation approaches have been proposed for chest X-rays (Sirazitdinov et al., 2019; Jaiswal et al., 2019), but none of these extend their work to the semi-supervised setting.

## 3. Methods

### 3.1. Setup

The multi-label semantic segmentation task is to classify each pixel of a chest X-ray image into zero or more of 10 possible pathologies: Airspace Opacity, Atelectasis, Cardiomegaly,

Consolidation, Edema, Enlarged Cardiomediastinum, Lung Lesion, Pleural Effusion, Pneumothorax, and Support Devices.

We utilize CheXpert (Irvin et al., 2019), an existing large dataset with 224,316 chest X-rays of 65240 patients. This dataset features image-level labels obtained using an automated labeler that detects the aforementioned pathologies from radiology reports. A subset of the dataset is hand-annotated by radiologists at the pixel level. In our work, we use a set of 200 radiologist-annotated chest X-rays to validate model performance of the weakly-supervised method. For the fully-supervised and semi-supervised methods, we use 150 of the radiologist-annotated labels as a train set and save 50 examples for a validation set. We selected this validation set to exclude scarce pathologies, as examples with those pathologies are most valuable in the training process. When evaluating performance on this validation set, we only look at the most common pathologies. For all methods, we use a test set of an additional 500 radiologist-annotated images.

Models are evaluated by their average performance on the semantic segmentation task across the ten pathologies of interest. For each of the pathologies, the IoU (Intersection-over-Union) score is computed. We report the mIoU (mean IoU) score, which is the average IoU score across all pathologies.

## 3.2. CheXseg

We develop *CheXseg*, a semi-supervised method for multi-pathology segmentation that leverages both the pixel-level expert annotations and the saliency maps generated by image classification models. In this method, a DenseNet121 (Huang et al., 2017) classification model, trained on the entire CheXpert train set, is first used to generate saliency maps using Grad-CAM (Selvaraju et al., 2017). This approach uses the classification model outputs to create a coarse localization map highlighting the image regions important for prediction. The saliency maps are further processed to create per-pixel segmentation masks, referred to as weak pseudo-labels, by using either a thresholding scheme or an Inter-Pixel Relation Network (IRNet) (Ahn et al., 2019). IRNet takes these generated CAMs and tries to improve the seeds by training two output branches, a displacement vector field and a class boundary map. Details about these methods are provided in Appendix A.

Once the pseudo-labels have been generated, we combine them with pixel-level expert annotations in a semi-supervised manner to train semantic segmentation models. Due to the scarcity of high-quality pixel-level expert annotations, we implement a sampling strategy of the different label types to allow for a high level of contribution from the radiologist annotations. For comparison, we train fully-supervised segmentation models (solely using pixel-level annotations) and weakly-supervised segmentation models (solely using pseudo-labels).

All our methods utilize DeepLabv3+ as the core semantic segmentation model (Chen et al., 2018). We experiment with various encoder initializations to transfer knowledge from the classification task to the segmentation task. Our experiments utilize a ResNet encoder architecture (He et al., 2016). Figure 1 shows a visual representation of the different supervision strategies used.

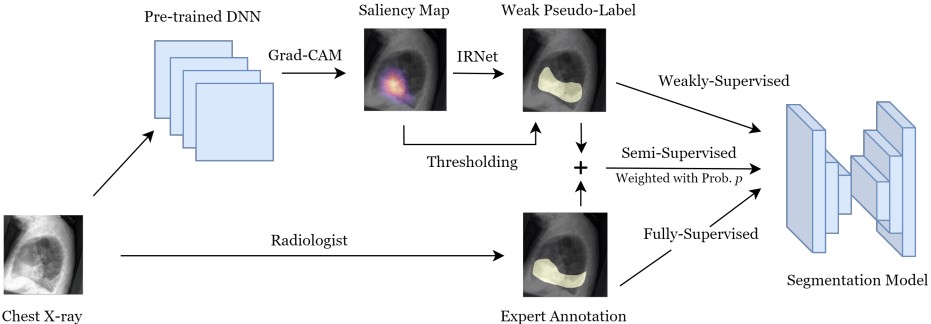

Figure 1: Workflows of the different methods analysed for chest X-ray segmentation

## 3.3. Training Details

Here we describe the training details of our classification and segmentation models. The IRNet training details are available in Appendix B.

### 3.3.1. CLASSIFICATION MODEL

The pre-trained classification model used for generating the CAMs from image-level labels is a DenseNet121. We use the Adam optimizer with default $\beta$-parameters of $\beta_1 = 0.9$, $\beta_2 = 0.999$ and learning rate $1 \times 10^{-4}$ which is fixed for the duration of the training. Batches are sampled using a fixed batch size of 16 images. We train for 3 epochs, saving checkpoints every 4800 iterations.

### 3.3.2. SEGMENTATION MODEL

The semantic segmentation model is trained using a class average dice loss and Adam optimizer. We use a learning rate of 0.001 when training a small amount of data and we decrease it to 0.0001 when training a large amount of data. We train on up to four Nvidia GTX 1070s using a batch size of 8.

## 4. Experiments

## 4.1. Combining Weak and Full Supervision

We investigate the segmentation performance of combining DNN-generated saliency maps and expert annotations with various sampling ratios. We use 100 saliency maps in combination with 200 annotated pixel-level labels and explore different weightings between the two types of labels. We vary the probability of selecting an expert annotation in a single batch during training, $p \in 0, 0.2, 0.4, 0.6, 0.8.0.85, 0.9, 1$. Thus, $p$ determines the expected fraction of images with expert annotations in a single batch. For each value of $p$, we perform three trials containing different sets of the 100 saliency maps. The results reported are the mIoU scores obtained by averaging across these 3 trials. The segmentation model is initialized with CheXpert encoder weights since it performs the best as observed in experiment 4.2.

We also compare the performance of this semi-supervised model with the weakly-supervised and fully-supervised models.

**Results** We find that for both Grad-CAM and IRNet, there is an inverted U-shape trend in performance as we increase $p$. There is a sharp increase in performance as $p$ increases from 0 (mIoU score of 0.156), and then the curves remain relatively flat before dropping off when $p = 1$. Specifically, $p = 0.9$ (*CheXseg*) gives the best mIoU performance of $0.270 \pm 0.00872$ and $0.267 \pm 0.00993$ (95% CI) for CAM and IRNet respectively. This high weighting of pixel-level labels takes advantage of the more accurate information encoded within these labels as compared to the saliency maps. The reduced performance for the fully-supervised case ($p = 1$, mIoU score of $0.246 \pm 0.01837$, 95% CI) is likely attributed to the weak labels no longer being utilized in training. The size of the train set shrinks, and the model does not benefit from the variation and the scale provided by the weak pseudo-labels. For the weakly-supervised case ($p = 0$, mIoU score of 0.156), the poor performance can be attributed to the absence of any pixel-level expert annotations to guide the model predictions. Detailed results are shown in Figure 2.

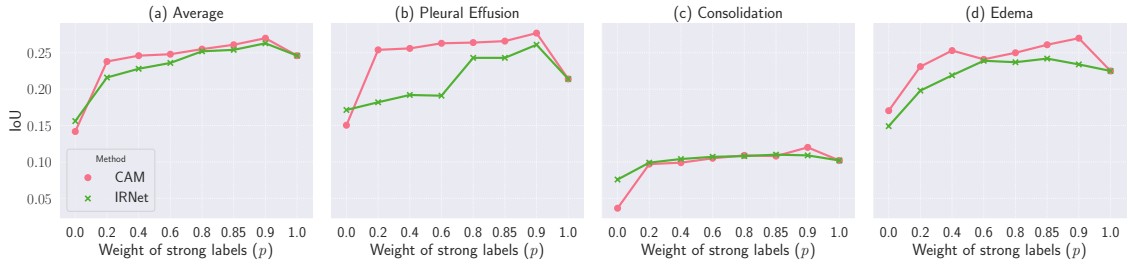

Figure 2: IoU scores of semi-supervised segmentation models (averaged across 3 trials) using either Grad-CAM or IRNet to generate weak labels. The DeepLabV3+ and ResNet18 setup is used with CheXpert encoder initializion. $p$ is the probability of selecting a pixel-level labeled training sample in the current batch. (a) is the average IoU score calculated across all pathologies. (b), (c), and (d) are IoU scores for Pleural, Consolidation, and Edema respectively. $p = 0$ represents the weakly-supervised case while $p = 1$ represents the fully-supervised case. Full results in Table 1.

## 4.2. Comparing Encoder Initializations

We investigate the impact of using various encoder initializations on segmentation performance. In the fully-supervised and weakly-supervised methods, we initialize the encoder weights to either a CheXpert classification model (Irvin et al., 2019), MoCo-CXR (Sowrirajan et al., 2020), ImageNet (Deng et al., 2009), or random.

**Results** We find that for all methods, the best models are initialized with CheXpert encoder weights. This may be expected since CheXpert weights are learned from supervised

learning on the same dataset that we use for segmentation. The models initialized with MoCo-CXR weights have similar performance to the models with ImageNet encoder initialization. Figure 3 shows the detailed results for fully-supervised and weakly-supervised encoder initializations.

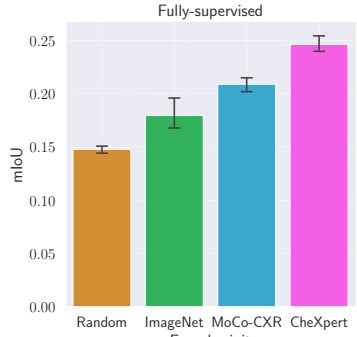 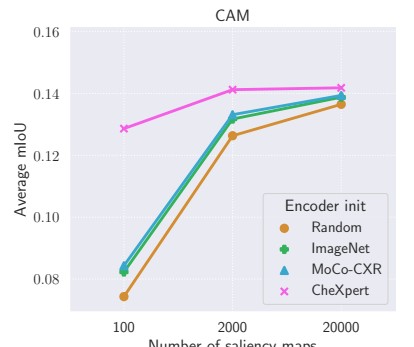 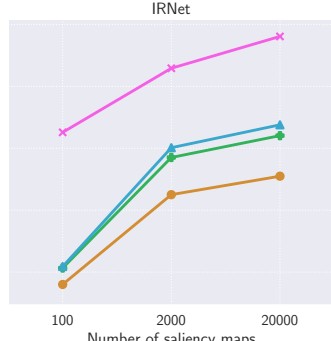

(a) mIoU of fully-supervised segmentation models with DeepLabV3+ and ResNet18 initialized with various weights. CheXpert encoder initialization results in best performance (average 0.246).

(b) mIoU of weakly-supervised segmentation models with DeepLabV3+ and ResNet18 setup using either Grad-CAM or IRNet pseudo-labels and various encoder initializations. IRNet outperforms CAMs when using CheXpert encoder initialization (0.156 vs 0.142), but underperforms when using other initializations (0.111 vs 0.136 with Random, 0.128 vs 0.139 with MoCo-CXR, and 0.124 vs 0.139 with ImageNet). Full comparisons with confidence intervals in Table 2.

Figure 3: Performance of fully-supervised and weakly-supervised methods

### 4.3. Comparison to Radiologists

Compared to our best weakly-supervised method, CheXseg reduces the overall performance (mIoU) gap with radiologists by 57.2%. CheXseg outperforms radiologists in terms of IoU score on the segmentation of Atelectasis (156% higher), Airspace Opacity (70% higher), and Pleural Effusion (30% higher), while performing worse on the remaining pathologies. Detailed results are shown in Figure 5.

### 5. Qualitative Results

Figure 4 shows the qualitative results for two pathologies - Cardiomegaly and Airspace Opeacity - for the best weakly-supervised (IRNet) and semi-supervised (CheXseg) methods. CheXseg gives better visualizations that are closer to ground truth as compared to the best weakly-supervised approach.

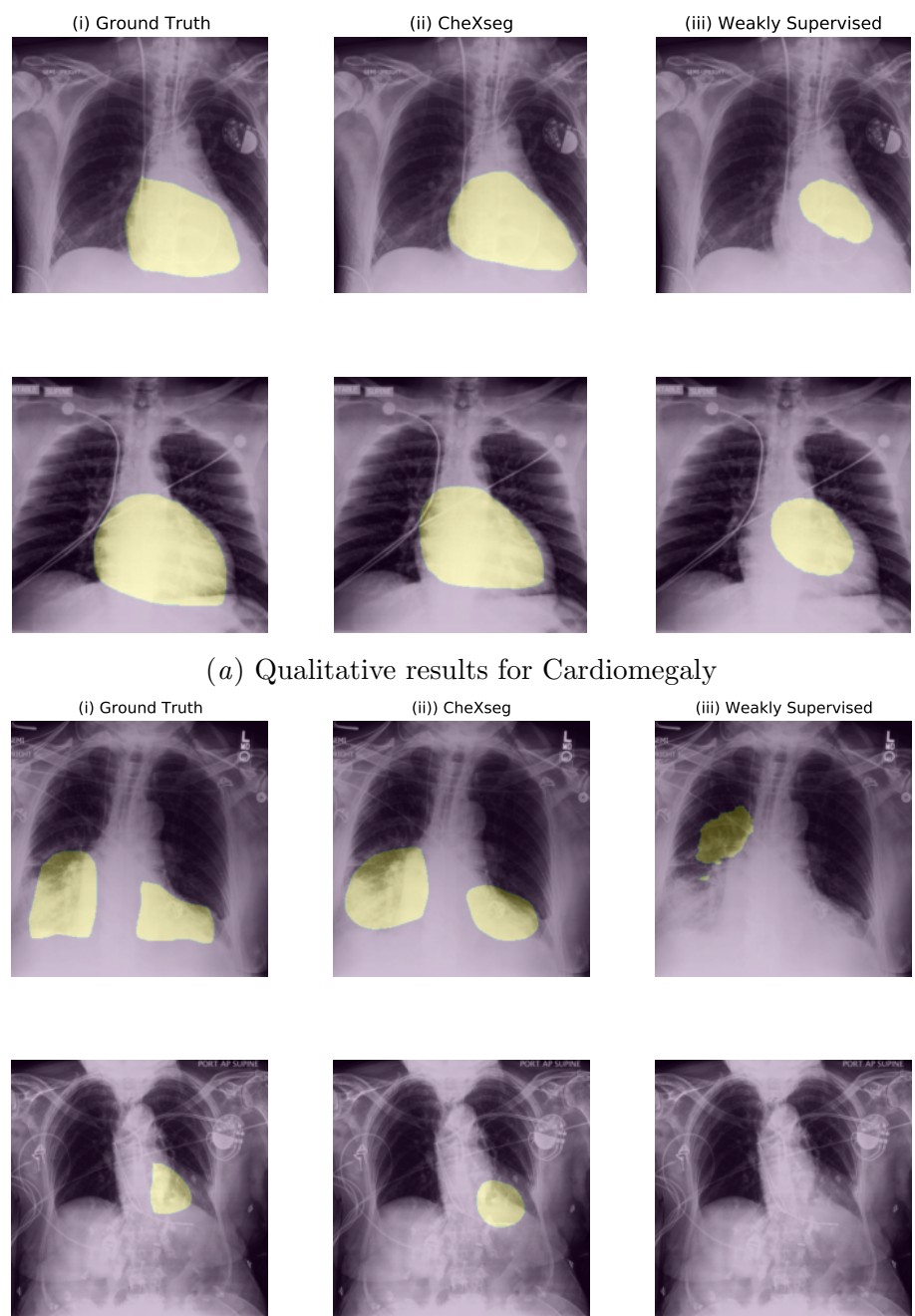

(a) Qualitative results for Cardiomegaly

(b) Qualitative results for Airspace Opacity

Figure 4: Qualitative Results for Cardomegaly and Airspace Opacity. Column (i) represents the Ground Truth segmentation map. Column (ii) represents the segmentation map obtained from CheXseg. Column (iii) represents the segmentation map obtained from using the best weakly supervised method (IRNet).

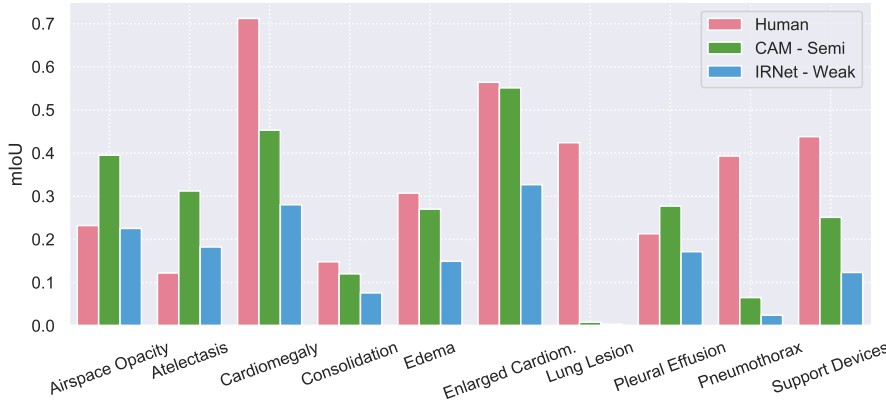

Figure 5: IoUs of radiologists, our best semi-supervised method (CheXseg), and our best weakly-supervised method. CheXseg uses Grad-CAM while the weakly-supervised method uses IRNet.

## 6. Discussion

In this work, we develop *CheXseg*, a semi-supervised method for multi-pathology segmentation that leverages the benefits of both available medical image classification models and expert pixel-level annotations.

*How does CheXseg performance compare to fully-supervised and weakly-supervised methods?* We find that with a weighted sampling of saliency maps and expert annotations, our proposed method outperforms both the fully-supervised and weakly-supervised methods alone. An expert annotation to saliency map sample ratio of 0.9 (CheXseg) gives the best mIoU score of 0.270, compared to 0.246 for fully-supervised and 0.156 for weakly-supervised. This suggests a tradeoff between emphasizing the accurate information of expert pixel-level annotations and incorporating additional but noisy cues from weak pseudo-labels.

*How do segmentation models initialized with CheXpert, ImageNet, MoCo-CXR, and random encoder weights compare?* We find that CheXpert encoder initialization achieves the highest performance, followed by self-supervised (MoCo-CXR) initialization and ImageNet initialization. Random encoder initialization has the worst performance, as no transfer learning occurs. Since CheXpert weights are pre-trained using image-level labels for the same tasks, it is expected that this knowledge transfers well to segmentation on the same dataset and the same set of tasks. Whereas classification models with MoCo-CXR encoder initialization have been found to outperform classification models with ImageNet encoder initialization (Sowrirajan et al., 2020), we find that the two initializations have approximately the same performance for segmentation.

In closing, our work proposes a simple semi-supervised method that combines the benefits of the wide availability of classification models with the quality of expert annotations. We expect our method may be broadly useful, able to lower the cost of development and improve the performance of medical image segmentation models.

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

## Appendix A. Methods for Generating Weak Pseudo-Labels

### A.1. Grad-CAM

Grad-CAM is used to obtain saliency maps from the model predictions to highlight the areas that the model focuses on during classification. It makes use of the gradients of the output classes flowing into the last convolutional layer to make low-resolution heatmaps ($3 \times 3$ in case of ResNets). Specifically, the gradients flowing back are global-average-pooled over the width and height dimensions (indexed by i and j respectively) to obtain the importance of the $k$th feature map for target class $c$, $\alpha_k^c$:

$$\alpha_k^c = \frac{1}{Z} \sum_i \sum_j \frac{\partial y^c}{A_{ij}^k} \tag{1}$$

Here, $y^c$ is the score (before softmax) of the class $c$, $Z$ is the normalization factor, and $A^k$ is the $k$th feature map activation.

After this, a weighted sum of the final feature maps followed by a ReLU is performed:

$$L_{Grad-CAM}^c = ReLU \left( \sum_k \alpha_k^c A^k \right) \tag{2}$$

A thresholding scheme is then used to convert the heatmap for each pathology into a segmentation map to use as pseudo-labels. The probability threshold is determined per pathology by maximizing the mIoU on the CheXpert train set.

### A.2. Inter-Pixel Relation Network (IRNet)

We follow the method IRNet, which takes the previously generated CAMs and tries to improve these seeds by training two output branches. The first branch predicts a displacement vector field in which each pixel is represented by a 2D vector pointing to the centroid of the instance that the pixel is a part of. This displacement field is converted to a class-agnostic instance map by grouping together pixels whose vectors point to the same centroid. The second branch is used to detect class boundaries by computing pairwise semantic affinities, which is a confidence score for class equivalence between a pair of pixels. The instance-wise CAMs obtained from the first branch are enhanced by propagating their attention scores to relevant areas using the computed affinities between neighboring pixels. Finally, pseudo-labels are generated independently for each pathology with positive pixels being the ones with higher class attention scores.

## Appendix B. Training Details

### B.1. IRNet

The two branches of IRNet share the same ResNet50 backbone and are jointly trained by minimizing the sum of three losses:

1. Loss for displacement field prediction, which consists of two losses:

(a) L1 loss between the image coordinate displacement between a pair of nearby foreground pixels $(i, j)$, denoted by $\hat{\delta}(i, j) = x_j - x_i$, and the displacement obtained from their vector fields, $\mathcal{D}$, denoted by $\delta(i, j) = \mathcal{D}(x_j) - \mathcal{D}(x_i)$:

$$\mathcal{L}_{\text{fg}}^{\mathcal{D}} = \frac{1}{|\mathcal{P}_{\text{fg}}^+|} \sum_{(i,j) \in \mathcal{P}_{\text{fg}}^+} \left| \hat{\delta}(i, j) - \delta(i, j) \right| \tag{3}$$

Here, $\mathcal{P}_{\text{fg}}^+$ refers to the set of neighboring foreground pixel pairs with the same pseudo label (pixels with attention scores larger than 0.3).

(b) Loss for background pixels as a normalized sum of their image coordinate displacements:

$$\mathcal{L}_{\text{bg}}^{\mathcal{D}} = \frac{1}{|\mathcal{P}_{\text{bg}}^+|} \sum_{(i,j) \in \mathcal{P}_{\text{bg}}^+} |\delta(i, j)| \tag{4}$$

Here, $\mathcal{P}_{\text{bg}}^+$ refers to the set of neighboring background pixel pairs with the same pseudo label (pixels with attention scores less than 0.05).

2. Loss for Class Boundary Detection, which makes use of the semantic affinities between a pair of pixels $x_i$ and $x_j$, $a_{ij}$:

$$a_{ij} = 1 - \max_{k \in \Pi_{ij}} \mathcal{B}(x_k) \tag{5}$$

Here, $\Pi_{ij}$ is a set of pixels on the line between $x_i$ and $x_j$ and $\mathcal{B} \in [0, 1]^{w \times h}$ is the output. Then, the loss is the cross-entropy loss between the binary affinity label, with value 1 for the same pseudo-class labels and 0 otherwise, and the predicted affinity of two pixels:

$$\mathcal{L}^{\mathcal{B}} = - \sum_{(i,j) \in \mathcal{P}_{\text{fg}}^+} \frac{\log(a_{ij})}{2|\mathcal{P}_{\text{fg}}^+|} - \sum_{(i,j) \in \mathcal{P}_{\text{bg}}^+} \frac{\log(a_{ij})}{2|\mathcal{P}_{\text{bg}}^+|} - \sum_{(i,j) \in \mathcal{P}^-} \frac{\log(1 - a_{ij})}{|\mathcal{P}^-|} \tag{6}$$

Here, $\mathcal{P}^-$ represents the set of pixel pairs with different pseudo labels.

The two branches are jointly trained by minimizing all three losses at the same time:

$$\mathcal{L} = \mathcal{L}_{\text{fg}}^{\mathcal{D}} + \mathcal{L}_{\text{bg}}^{\mathcal{D}} + \mathcal{L}^{\mathcal{B}} \tag{7}$$

The model is trained with stochastic gradient descent using a learning rate of 0.1 with polynomial decay and a batch size of 16. The segmentation maps created are then used in the semantic segmentation model as weak pseudo-labels.

# Appendix C. Semi-Supervised Results

Table 1: IoU scores for semi-supervised segmentation using IRNet and CAMs by weighting of pixel-level and weak pseudo-labels. $p$ represents the probability of picking a expert annotated training example in the current batch.

| Task | Method | $p{=}0$ | $p{=}0.2$ | $p{=}0.4$ | $p{=}0.6$ | $p{=}0.8$ | $p{=}0.85$ | $p{=}0.9$ | $p{=}1$ |
|---|---|---|---|---|---|---|---|---|---|
| Mean IoU | CAM | **0.142** | **0.238** | **0.246** | **0.248** | **0.255** | **0.261** | **0.270** | 0.246 |
| | IRNet | 0.156 | 0.216 | 0.228 | 0.236 | 0.252 | 0.254 | 0.267 | |
| Airspace Opacity | CAM | 0.161 | **0.364** | **0.381** | **0.382** | **0.376** | 0.376 | 0.398 | 0.388 |
| | IRNet | **0.225** | 0.334 | 0.355 | 0.362 | 0.374 | **0.377** | **0.403** | |
| Atelectasis | CAM | **0.099** | **0.299** | **0.295** | **0.297** | 0.306 | 0.299 | 0.310 | 0.307 |
| | IRNet | 0.182 | 0.269 | 0.278 | 0.287 | **0.316** | **0.308** | **0.323** | |
| Cardiomegaly | CAM | **0.326** | **0.420** | **0.438** | **0.434** | **0.440** | **0.442** | 0.450 | 0.445 |
| | IRNet | 0.280 | 0.393 | 0.401 | 0.425 | 0.428 | 0.440 | **0.461** | |
| Consolidation | CAM | **0.056** | 0.097 | 0.099 | 0.105 | **0.109** | 0.108 | **0.114** | 0.103 |
| | IRNet | 0.076 | **0.099** | **0.104** | **0.107** | 0.108 | **0.110** | 0.110 | |
| Edema | CAM | **0.170** | **0.231** | **0.253** | **0.241** | **0.250** | **0.261** | **0.270** | 0.225 |
| | IRNet | 0.149 | 0.198 | 0.219 | 0.239 | 0.237 | 0.242 | 0.257 | |
| Enlarged Cardiomediastinum | CAM | 0.236 | **0.516** | **0.494** | **0.494** | 0.514 | 0.526 | **0.543** | 0.549 |
| | IRNet | **0.327** | 0.466 | 0.480 | 0.489 | 0.528 | 0.531 | 0.535 | |
| Lung Lesion | CAM | 0.002 | **0.004** | **0.003** | **0.003** | 0.005 | 0.008 | **0.008** | 0.002 |
| | IRNet | **0.003** | 0.002 | 0.002 | 0.003 | **0.014** | **0.014** | 0.007 | |
| Pleural Effusion | CAM | **0.150** | **0.254** | **0.256** | **0.263** | **0.264** | **0.266** | 0.272 | 0.214 |
| | IRNet | 0.171 | 0.182 | 0.192 | 0.191 | 0.243 | 0.243 | **0.273** | |
| Pneumothorax | CAM | **0.057** | **0.021** | 0.025 | **0.035** | **0.044** | **0.074** | **0.077** | 0.017 |
| | IRNet | 0.024 | 0.018 | **0.029** | 0.033 | 0.040 | 0.050 | 0.053 | |
| Support Devices | CAM | **0.159** | **0.203** | **0.223** | **0.231** | **0.241** | **0.249** | **0.257** | 0.246 |
| | IRNet | 0.123 | 0.202 | 0.217 | 0.226 | 0.229 | 0.221 | 0.248 | |

# Appendix D. Weakly-Supervised Results

Table 2: IoU scores of weakly-supervised segmentation models using either CAM or IRNet pseudo-labels and varying encoder architectures and train set sizes. Confidence intervals are calculated using $\alpha = 0.05$.

| Method | Encoder Architecture | Train Set Size | Test mIoU |
|---|---|---|---|
| CAM | ResNet18 | 100 | $0.074 \pm 0.00783$ |
| | | 2000 | $0.126 \pm 0.00275$ |
| | | 20000 | $0.136 \pm 0.00990$ |
| | ResNet18-ImageNet | 100 | $0.082 \pm 0.00396$ |
| | | 2000 | $0.132 \pm 0.00687$ |
| | | 20000 | $0.139 \pm 0.00095$ |
| | ResNet18-MoCo-CXR | 100 | $0.084 \pm 0.00238$ |
| | | 2000 | $0.133 \pm 0.00002$ |
| | | 20000 | $0.139 \pm 0.00076$ |
| | ResNet18-CheXpert | 100 | $0.129 \pm 0.00036$ |
| | | 2000 | $0.141 \pm 0.00003$ |
| | | 20000 | $0.142 \pm 0.00095$ |
| IRNet | ResNet18 | 100 | $0.076 \pm 0.00133$ |
| | | 2000 | $0.105 \pm 0.00292$ |
| | | 20000 | $0.111 \pm 0.00103$ |
| | ResNet18-ImageNet | 100 | $0.081 \pm 0.00245$ |
| | | 2000 | $0.117 \pm 0.00576$ |
| | | 20000 | $0.124 \pm 0.00001$ |
| | ResNet18-MoCo-CXR | 100 | $0.082 \pm 0.00007$ |
| | | 2000 | $0.120 \pm 0.00010$ |
| | | 20000 | $0.128 \pm 0.00001$ |
| | ResNet18-CheXpert | 100 | $0.125 \pm 0.00031$ |
| | | 2000 | $0.146 \pm 0.00002$ |
| | | 20000 | $0.156 \pm 0.00000$ |

