# OpenReview forum: "CheXseg: Combining Expert Annotations with DNN-generated Saliency Maps for X-ray Segmentation"
_MIDL.io/2021/Conference — MIDL 2021_

### Official Review · AnonReviewer2 · 2021-03-06

**Confidence:** 4
**Preliminary Rating:** 3
**Recommendation:** Poster

**Summary:**

This paper investigates the use of image-level labels to pretrain a DNN in order to obtain weak-labels from class saliency maps for semi-supervised (SS) pixel-level segmentation. The authors compare their SS method against a fully supervised method and a weakly supervised method, showing that a small fraction of unabelled data can improve against the fully supervised baseline. The authors also compare different initialisations for the segmentation network encoder to further validate.

**Strengths:**

1. The results of this paper are promising as it clearly shows how global image labels can be leveraged in radiography to help train more accurate segmentation networks when manual ground-truth data is expensive to acquire.

2. The authors provide enough experiments to show how adding weak labels from a pre-trained DNN on global image labels helps improve on a fully-supervised model.


**Weaknesses:**

[1] Main weakness of the paper relates to the lack of baselines. Despite the authors comparing against fully supervised and a weakly supervised model, I would have like to see how the model performs against similar methods that use self-supervision for weak supervsion in a semi-supervised set up. The authors bring up related work in Section 2.1. Was it not possible to compare? How about other methods such as https://arxiv.org/pdf/2004.04581.pdf etc.?


[2] I am also uncertain on the novelty of the method. The idea of using grad-CAM to generate saliency maps as weak labels is not new as reported by the authors in 2.1. The authors do not propose a novel methodology based on this for semi-supervision as the parameter *p* only governs the sampling of labelled/unlabelled data per batch. I am happy to be challenged on this. However, from my perspective, the novelty of this paper is in the application towards chest radiography, not its method.

**Deanonymize Review:**

no

**Detailed Comments:**

1. Could the authors clarify on the exact number of hand-annotated pixel maps in the dataset - is it 700 out of 224,316 xrays?

2. When the DNN for generating saliency maps is pre-trained, do you include the scans which have ground-truth labels and which constitute the supervised set?

3. Why is the human performance so bad for airspace opacity, atelectasis and pleural effusion?

4. I would like to see all baseline models included in Figure 5. Do radiologists always beat the fully supervised model for instance.

5. Examples of the segmentation outputs and saliency maps compared to expert annotation would be favourable. Do the errors correspond to issues in the saliency maps? Why does the model perform so well for certain labels but poorly for others? Is this related to the weak supervision for unlabelled cases?

6. Given there is so much unlabelled data to be used, is it possible to consider an active learning set up with the pre-trained DNN for getting saliency maps?

**Justification Of The Preliminary Rating:**

My preliminary rating of weak accept is based on the fact it is generally well written, the conclusion is corroborated by the results and the proposed method clearly improves on the upper-bound that is a fully supervised algorithm. Further, showing that global image labels can be used for self-supervision to generate weak labels in semi-supervised learning is important for medical imaging where ground truth labels are laborious and expensive to acquire.

I have only given weak as I believe the novelty of the method to be limited, a lack of comparison with SOTA baselines and more detailed analysis of the results could be present.

**Paper Type:**

both

**Questions To Address In The Rebuttal:**

Main points based in Weaknesses such as novelty and lack of baselines to compare against. Otherwise, some of my questions in detailed comments to improve the manuscript.

**Special Issue:**

yes

---

> ### Author Response · Authors · 2021-03-18
> **We thank the reviewer for their encouraging and constructive comments.**
>
> We have included point-by-point responses to the reviewer’s questions below.
>
> 1. *Baselines:* In our revised manuscript, we can include a baseline which uses self-supervision in a semi-supervised setup. We chose to use a GradCAM and IRNet setup for weak supervision because it had the best performance on natural images and was already proven to perform well on a medical imaging task. One thing to note about self-supervision for weak supervision is that many methods rely on accurate transformations of images. In our setting, such transformations may make incorrect assumptions about the data.
>
> 2. *Novelty:* We believe that our sampling strategy is a novel approach in semi-supervised learning, especially in its application towards chest radiography. Previous work, such as SGAN which we reference in 2.1, replaces a subset of pseudo-labels with ground-truth annotations without any sampling. With our data, this semi-supervised approach does not perform well since there are so few annotated images to work with that the saliency maps introduce too much noise. Our sampling strategy allows models to improve upon fully-supervised performance even when the number of annotations is extremely small. In the revised manuscript, we will better explain the novelty of our method and how it differs from previous work.
>
>
> Detailed Comments:
>
> *Could the authors clarify on the exact number of hand-annotated pixel maps in the dataset?*
>
> Yes there are 700 hand-annotated pixel maps in the dataset. 200 of these are used as the training set for the fully-supervised method and 500 other pixel maps are used as the test set
>
> *When the DNN for generating saliency maps is pre-trained, do you include the scans which have ground-truth labels and which constitute the supervised set?*
>
> No, the DNN pre-training does not include any scans which are present in the supervised set. The DNN is trained using the train set of the CheXpert dataset along with the classification labels.
>
> *Why is the human performance so bad for airspace opacity, atelectasis and pleural effusion? Do radiologists always beat the fully supervised model for instance?*
>
> One potential explanation for poor performance is that pathologies can be hard to distinguish, especially without clinical context. Also, radiologists use a certain amount of clinical discretion when defining the boundaries of a pathology on a chest X-ray. There can also be institutional and geographic differences in how radiologists are taught to recognize pathologies, and studies have shown that there can be high interobserver variability in the interpretation of CXRs [1,2,3].
>
> There are also certain pathological characteristics which can affect human performance, such as the number of instances for a given pathology, pathology area with respect to the whole X-ray area, elongation, and irrectangularity.
>
> We will make sure to include all baseline models in Figure 5 in the revised manuscript. The fully supervised model outperforms radiologists on 3 of the 10 pathologies: Airspace Opacity, Atelectasis, and Pleural Effusion.
>
> *Examples of the segmentation outputs and saliency maps compared to expert annotation would be favourable. Do the errors correspond to issues in the saliency maps? Why does the model perform so well for certain labels but poorly for others? Is this related to the weak supervision for unlabelled cases?*
>
> We will add qualitative examples in the revised manuscript. The main reason for the model having varied performance on different labels can be attributed to the skewed nature of the number of expert annotations for the labels. There are some pathologies like Lung Lesion and Pneumothorax which only have 1-2 images with expert annotations in the train set. This number might be too less for the model to learn anything substantial, resulting in lower performance for these labels. This pattern seems to be similar to the one seen in the weak supervision case - similar pathologies perform poorly for the weakly-supervised methods.
>
> *Given there is so much unlabelled data to be used, is it possible to consider an active learning set up with the pre-trained DNN for getting saliency maps?*
>
> Yes, this is definitely a consideration, but we have not explored an active learning setup in this paper. This is because we don’t use the full train set provided for any of the segmentation experiments. The only time the entire CheXpert train set is used is for training the DNN used for generating saliency maps. We conjecture that 10% of CheXpert, or roughly 20,000 images, is a sufficiently representative sample of the dataset.
>
> [1] Melbye, H. & Dale, K. Interobserver Variability in the Radiographic Diagnosis of 666 Adult Outpatient Pneumonia. Acta Radiol. 33, 79–81 (1992).
> [2] Herman, P. G. et al. Disagreements in Chest Roentgen Interpretation. CHEST 668 68, 278–282 (1975). 669
> [3] Albaum, M. N. et al. Interobserver Reliability of the Chest Radiograph in 670 Community-Acquired Pneumonia. CHEST 110, 343–350 (1996).

---

### Official Review · AnonReviewer3 · 2021-03-08

**Confidence:** 3
**Preliminary Rating:** 2
**Final Rating:** 3

**Summary:**

The paper proposes a weakly- and semi-supervised learning setup for learning a model for segmentation of pathologies on chest X-rays. The approach is evaluated on the CheXpert dataset and shows that the use of weak semi-supervised learning improves greatly upon semi-supervised learning or supervised learning only. For this the paper compares the use of saliency-based pseudo-labels, self-distillation for semi-supervised learning and fully-supervised models.

**Strengths:**

The paper tackles an interesting problem in how to train segmentation networks with limited pixel-level annotations but a vast amount of image-level annotations. It builds on top of recent advances in weakly supervised and semi-supervised learning and provides a reasonable set of evaluations across different parameters that influence the performance of this approach.

**Weaknesses:**

The paper is mainly an application paper with limited methodological novelty but seems to lack some experimental rigour that let me question the validity of some of the drawn conclusions about CheXseg being the best performing approach:
- Has a validation set been used or how where the parameters tuned?
- Why is only 10% of data used for the self-distillation experiment as this seems to improve performance and doesn't require annotations?
- How was the number of 100 used saliency maps for CheXseg chosen? Again, it should be easy to increase this number and would be beneficial to make statements about the trends or at least tune this on a validation set.
- Why are the values for p=0 in table 1 not consistent with table 2?

If those questions can be reliably addressed I'm happy to revise my recommendation.

**Deanonymize Review:**

no

**Detailed Comments:**

- I believe the DenseNet citation might be wrong. I think you might mean [1].
- In 3.2. you mention that IRNet improves the seeds but at time it is not clear what a seed is.
- How important is the performance and the architecture of the classification network to produce useful saliency maps?
- Have you thought about combining pseudo-labels and the distillation method?
- Did you experiment with different numbers of used saliency maps in 4.1? How important is the ratio between number of saliency maps and expert annotations?
- For 4.2. it is mentioned that the student model is trained using 1% or 10% of the data. Why aren't bigger settings evaluated? Also, I assume this refers to the full CheXpert dataset?
- For 4.4 the performance of radiologists - what do you assume as gold-standard ground truth if not the radiologists annotations? I assume it's the majority vote of the different annotators - then it should be possible to add error bars to Fig. 5. Also, it would be helpful to add an average category to Fig. 5. Further, how do you calculate the ~72% improvement compared to weakly supervised methods?
- It seems that the experiments are performed using a train and test split only - was there a validation set involved? How was this chosen?
- For Table 2, should the performance for ResNet18-CheXpert model with 100 training samples be the same as the p=0 model in table 1? What's the difference there?
- Does the CheXpert dataset already come with segmentation annotations? I had a quick glance at the paper and page and couldn't find a mention of this.
- What's the distribution of pathologies in the annotated segmentations? Is there a imbalance that would have an impact of the student-teacher training setup?
- (Very personal opinion): I would have preferred to see most of the results in table format or at least have them available in tables as it's easier to compare the exact values.

[1] Huang, Gao, et al. "Densely connected convolutional networks." Proceedings of the IEEE conference on computer vision and pattern recognition. 2017.

**Final Rating Justification:**

I thank the authors for their comments - especially the revised experiments which alleviated some of my concerns about the results.  I believe the paper proposes a simple yet effective method for weakly supervised segmentation. The novelty of the method is limited but the paper offers an interesting combination of methodological contributions and validation on the chest x-ray dataset.

**Justification Of The Preliminary Rating:**

The paper introduces an interesting approach for weakly supervised segmentation of chest X ray pathologies, even though, having little novelty. I am insecure about some of the experimental setup and execution and hope that the authors can clarify this for me to revise my rating.

**Paper Type:**

validation/application paper

**Questions To Address In The Rebuttal:**

See weaknesses.

**Special Issue:**

no

---

> ### Author Response · Authors · 2021-03-18
> **We would like to thank the reviewer for their constructive comments.**
>
> We have included point-by-point responses to the reviewer’s questions below.
>
> *Has a validation set been used or how were the parameters tuned?*
>
> For the weakly-supervised method, we use the set of 200 radiologist-annotated chest X-rays to validate model performance and save the best checkpoint, and we do final testing on the test set of 500 radiologist-annotated images. Previously, for the fully-supervised and semi-supervised methods (which use the set of 200 annotated X-rays for training), we saved checkpoints along the way during training, evaluated those checkpoints on the test set, and reported the performance of the best checkpoint. Since then, we have updated and strengthened our experimental setup, creating a validation set of 40 images from the set of 200 annotated X-rays. We selected the validation set to not include the scarce pathologies, as those examples are most valuable in the training process. To validate model performance and save the best checkpoint, we measure the ious of the most common pathologies. We only use the test set for final testing.
>
> As we are solely comparing performance within our methods, all our findings and claims are unchanged. The only difference we encounter is all fully-supervised and semi-supervised methods’ performance drops slightly. This is due to the fact that we now train on a smaller amount of radiologist-annotated images, and there is no model selection on the test set. However, both methods still strongly outperform the weakly-supervised method, and an expert annotation sampling rate of 0.9 results in best performance. Here are updated results for fully-supervised and semi-supervised methods that will go in the revised manuscript.
>
> Fully-supervised:
>
> |Encoder init|Random|ImageNet|MoCo-CXR|CheXpert|
> |---|---|---|---|---|
> |mIoU|0.148 ± 0.0037|0.180 ± 0.0166|0.209 ± 0.0074|0.246 ± 0.0084|
>
> Semi-supervised:
>
> |*p* value|*p*=.7|*p*=.8|*p*=.9|*p*=1|
> |---|---|---|---|---|
> |mIoU|0.231 ± 0.0053|0.251 ± 0.0205|**0.266  ± 0.0187**|0.246 ± 0.0084|
>
> Thank you for bringing this up and pushing us to strengthen our experimental setup and execution.
>
> *Why is only 10% of data used for the self-distillation experiment as this seems to improve performance and doesn't require annotations?*
>
> During experimentation, we found that using above 10% of the CheXpert dataset resulted in no performance improvement. We conjecture that 10% of CheXpert, or roughly 20,000 images, is a sufficiently representative sample of the dataset. We will include the 100% data distillation experimental results in the revised manuscript, as well as suggested reasoning for the results.
>
> *How was the number of 100 used saliency maps for CheXseg chosen? Again, it should be easy to increase this number and would be beneficial to make statements about the trends or at least tune this on a validation set.*
>
> We experimented with using different amounts of saliency maps, specifically powers of 10 with 100 being the smallest. We found that using more than 100 saliency maps did not improve performance. We will include experimental results for this in the revised manuscript.
>
> *Why are the values for p=0 in table 1 not consistent with table 2?*
>
> Thanks for pointing this out; we have revised table 1 to have correct values for p=0.
>
> With the following modifications, we hope that the reviewer can revise their rating.
>
> Detailed comments:
>
> Due to the character limit, we are not able to answer all the detailed comments, but we thank the review for all the feedback and catching errors that we have fixed for the revised manuscript.
>
> *Seed clarification*: The seeds are the Class Activation Maps generated by applying Grad-CAM to the outputs of the classification model. IRNet is used to improve upon these CAMs by training a displacement vector field and detecting class boundaries. In our revised manuscript, we will give more clarification on what the seeds are. For the classification network, we use the architecture proposed in CheXpert. We observe that higher performance of the classification network on the test set results in better saliency maps obtained.
>
> *Combining pseudo-labels and the distillation method?*: Currently, we are only using the distillation method as a semi-supervised baseline to compare against our proposed method, but we can certainly combine the approaches.
>
> *Gold standard ground truth*: We use the radiologist annotations as the ground truth to measure model segmentation performance, and we use additional benchmark segmentations to measure expert segmentation performance.
>
> *CheXpert segmentation annotations*: These annotations will soon be released, and so will our code.
>
> *Distribution of pathologies in the annotated segmentations*: Here is the distribution of pathologies in the set of 200 annotated segmentations:
>
> 'Cardiomegaly': 68, 'Enlarged Cardiomediastinum': 109, 'Edema': 45, 'Support Devices': 107, 'Atelectasis': 80, 'Pneumothorax': 8, 'Pleural Effusion': 67, 'Consolidation': 33, 'Lung Lesion': 1

---

### Official Review · AnonReviewer1 · 2021-03-09

**Confidence:** 5
**Preliminary Rating:** 4
**Recommendation:** Best Paper Award, Oral
**Final Rating:** 4

**Summary:**

The work proposes a semi-supervised approach for semantic segmentation of Chest X ray into 10 pathologies. As indicated in this work, annotating each and every pixel for training supervised semantic segmentation approaches limits its use to practical applications.  The proposed work uses the classification level annotation (Grad CAM) to generate the weak pseudo labels, along with the limited expert annotation. The result presented on CheXpert dataset shows that the semi-supervised approach outperforms the fully supervised and weakly supervised approaches.

**Strengths:**

In my opinion, these are strong points of the proposed work:

1) Methodology: The fusion of weakly supervised with a fully supervised approach to address the challenge posed by the availability of annotated data is very interesting.  This simple idea opens up the possibility of deploying CNN-based semi-supervised approach for semantic segmentation of Chest X ray in hospitals, especially in resource-constrained settings.

2) Results: The proposed approach has been evaluated on CheXpert dataset and thorough experimental results are presented in the work. Above all, the discussion clearly summarizes the main observations of the results presented (comparison with fully supervised with weakly supervised, performance analysis of semi-supervised distillation, and various encoder initialization).

3) Presentation: The paper is very well presented and easy to follow. The questions that developed while reading the paper, were already answered in the manuscript!

**Weaknesses:**

As explained in the strengths of the paper, it is very difficult to identify the weaknesses of this work. The questions that developed while reading the paper, were already answered in the manuscript!

However, I would be interested in the details of the thresholding operation utilized in creating the weak pseudo labels from the saliency map. Was the threshold experimentally determined?

If I am not wrong, the pseudo labels were created by either using thresholding operation or IRNet. On what basis, this choice was made?

**Deanonymize Review:**

no

**Final Rating Justification:**

Thanks for the reply. I am satisfied with the comments. However, it would be better to include these two points in the final camera ready version:

1) Which thresholding approach (Otsu?) was utilised and why?
2) A brief summary of why IRNet was selected for generating pseudo labels (even if it is obvious)...

**Justification Of The Preliminary Rating:**

The paper proposes a simple solution to address one of the major challenges of the supervised deep learning approach.  The fusion of weakly supervised and fully supervised network presented in this work shows promising results and would be immensely useful for radiologists.

**Paper Type:**

methodological development

**Special Issue:**

yes

---

> ### Author Response · Authors · 2021-03-18
> **We thank the reviewer for their exceedingly encouraging comments.**
>
> We have included point-by-point responses to the reviewer’s questions below.
>
> *I would be interested in the details of the thresholding operation utilized in creating the weak pseudo labels from the saliency map. Was the threshold experimentally determined?*
>
> We set a threshold on the model output probability; above that threshold we normalized the heatmap and used the standard cv2 package to threshold; below the probability threshold we just say it’s all 0s. The probability threshold is determined per class by maximizing mIoU on the validation set with 200 images.
>
> *If I am not wrong, the pseudo labels were created by either using thresholding operation or IRNet. On what basis, this choice was made?*
>
> Yes, that is correct - either the thresholding or IRNet was used to generate the pseudo labels. Both of these methods were tested independently and it was observed that for the weakly-supervised method, IRNet performed better than the thresholding in terms of the mIoU score but for the semi-supervised scenario, the thresholding performed better.

---

### Official Review · ~Pedro_M._Gordaliza1 · 2021-03-09

**Confidence:** 4
**Preliminary Rating:** 2

**Summary:**

The authors propose a method for automatic segmentation of the main manifestations (10 labels) commonly found in pathological lungs when using Chest X-Ray.  To mitigate the usual lack of expert-annotated images, especially in the case of segmentation, the authors propose to train a segmentation model using semi-supervision by combining annotations from experts and saliency-maps (weak-labels) in different proportions, experiments ranging from not using expert annotations at all (weak-supervision) to using only expert annotations (fully supervised). The saliency-maps are obtained from the well-known classification-model CheXpert in two ways: 1) By the standard Grad-CAM (Selvaraju et al., 2017) with thresholding and 2) IRNet which intends to provide better pseudo-labels
Besides, the authors propose a self-destilation segmentation model where the encoder is initialised with the CheXpert weights.


**Strengths:**

- All approaches that propose the use of unsupervised or semi-supervised learning to exploit the information contained in large databases are of great interest and fundamental for the immediate future in the field of medical imaging.

- Experiments varying the ratio between weak labels and expert annotations are essential to characterize the usefulness or otherwise of this type of method.

- The authors propose and measure the performance of two SOTA methods such as Grad-cam and IRNet for the generation of weak-labels.

- The authors experiment and compare different initializations, which is a must to provide robust results on the performance of the method.

**Weaknesses:**

- Recent literature [1-3] indicates that the use of semi-supervised methods in segmentation is not the most appropriate strategy. Very briefly and without formalism, one could say that the architectures cannot replicate the segmentation mechanisms (P(Y|X)) synthesised in the expert annotations. In [1] it is shown that those studies that report a better segmentation using SSL usually have some kind of bias, also in this same work a framework is proposed to reduce them as much as possible, however, most of the recommendations have not been implemented in this work.
In this context, it is well-known that saliency maps usually highlight regions outside the lungs[4] as any kind of marker in the image which correlates with the label. This fact implies huge generalization errors that had not been taken into account in this work.

- The code is not available.

- Most of the results are given without error measures

- The comparison and results for the distillation model are unclear

- Just quantitative results are provided. Since the authors are presenting a segmentation paper would be highly recommendable to include images of the resultant segmentation.





[1] Oliver, A., Odena, A., Raffel, C., Cubuk, E. D., & Goodfellow, I. J. (2018). Realistic Evaluation of Deep Semi-Supervised Learning Algorithms. Advances in Neural Information Processing Systems, 2018-December, 3235–3246. Retrieved from http://arxiv.org/abs/1804.09170
[2] Arjovsky, M., Bottou, L., Gulrajani, I., & Lopez-Paz, D. (2020). Invariant Risk Minimization. http://arxiv.org/abs/1907.02893
[3] Castro, D. C., Walker, I., & Glocker, B. (2020). Causality matters in medical imaging. Nature Communications, 11(1), 1–10. https://doi.org/10.1038/s41467-020-17478-w
[4] Degrave, A. J., Janizek, J. D., Lee, S.-I., & Allen, P. G. (2020). AI for radiographic COVID-19 detection selects shortcuts over signal. MedRxiv, 2020.09.13.20193565. https://doi.org/10.1101/2020.09.13.20193565

**Deanonymize Review:**

yes

**Justification Of The Preliminary Rating:**

The idea is interesting but the work lacks some details that would give it more packaging and maturity. The conclusions drawn are not clear if the results presented are taken into account. However, with some corrections the work could be presented as a poster.

**Paper Type:**

methodological development

**Questions To Address In The Rebuttal:**

- Always include error measures with the results (even with 3 draws...). This work claims a better performance using a SSL framework than a fully-supervised one, however, the mIOUs are so small and close that the results could be perfectly due exogenous uncontrolled variables   so at least, could you please include a proper statistical analysis?

- It would be very interesting to see some comparison with SOTA methods. Considering the input data available, an interesting case would be to use the framework proposed in [5] using saliency maps and expert annotations as different raters.

- There are huge differences between the segmentation results obtained with the proposed methods and the results of the radiologists depending on the type of manifestation, why?

- “We experiment with various encoder initializations to transfer knowledge from the classification task to the segmentation task. Our experiments utilize a ResNet encoder architecture...” It is difficult to identify the different configurations of the experiments in the text, could you include a table summarizing the different set-ups of the experiments?

- Figure 1 can be confusing, it gives the impression that different annotations make up the input to the segmentation model rather than the image itself. Could it be slightly modified?

- Could you please include some details about the annotated images? i.e: number of radiologists, experience, etc.

- Why is the distillation model based on fully-supervised training if in the previous experiment the results are better with p=0.9?

- Could you please expand on this "MoCo-CXR weights have similar performance to the models with ImageNet encoder initialization"? This is strange given that MoCo-CXR should "a priori" initialize better given the similarity of the information used.




[5] Kohl, S. A. A., Romera-Paredes, B., Meyer, C., De Fauw, J., Ledsam, J. R., Maier-Hein, K. H., … Ronneberger, O. (2018). A Probabilistic U-Net for Segmentation of Ambiguous Images. Advances in Neural Information Processing Systems, 2018-December, 6965–6975. Retrieved from http://arxiv.org/abs/1806.05034



**Special Issue:**

no

---

> ### Author Response · Authors · 2021-03-18
> **We thank the reviewer for their constructive comments.**
>
> We have included point-by-point responses to the reviewer’s weaknesses and questions below.
> 1. *Semi-supervised learning and generalization.*
> We strongly believe that a semi-supervised approach to segmentation is most appropriate for our setting and that our work does not rely on biases in the data. Our setup differs from the mentioned works in that our weak-supervision source has classification labels, just not expert segmentation masks. The classification labels still provide supervision for training that can be utilized to improve model performance. In addition, our setting of segmenting pathologies cannot rely on the mentioned biases to perform well and must learn the relationship between the image and corresponding radiologist-annotated segmentations. The model cannot use cheats like in [4], since relying on markers in regions outside the lungs would cause the model to highlight those regions, therefore resulting in poor segmentation performance.
>
> 2. *The code is not available.*
> The code will certainly be made available. We have been preparing the code for release, and are in the process of testing our installation and usage instructions.
>
> 3. *Most of the results are given without error measures:*
> We will make sure to include confidence intervals for all of our results in the revised manuscript.
>
> 4. *The comparison and results for the distillation model are unclear.*
> We use distillation as a semi-supervised comparison for our proposed method, CheXseg. Instead of using saliency maps as pseudo-labels, the distillation method uses a fully-supervised teacher model’s predictions on unlabeled images as pseudo-labels. We find that the distilled students have higher segmentation performance than their fully-supervised teacher models. Yet, this distillation method is still outperformed by CheXseg. In the revised manuscript, we will try to be more clear about the comparisons of this method and results for distillation.
>
> 5. *Just quantitative results are provided. Since the authors are presenting a segmentation paper would be highly recommendable to include images of the resultant segmentation.*
> We will make sure to add images as qualitative results in the revised manuscript.
>
> Questions to address -
> *Bullet 1:*
> We will include confidence intervals for all of our results in the revised manuscript. We will also include statistical analyses between the performances of the various methods.
>
> *Bullet 2:*
> For our revised manuscript, we are working on comparing our method with baselines that achieve SOTA performance on other datasets, such as https://arxiv.org/abs/2012.05007. We agree that it would be interesting to see this comparison.
>
> *Bullet 3:*
> The differences in performance may be attributed to the amount of annotated segmentations available for different pathologies. The pathologies that our proposed methods have the worst relative performance on -- Pneumothorax and Lung Lesion -- have just a few images with annotated segmentations in the train set. This number may be too small for the model to learn anything substantial, resulting in lower performance for these labels. This pattern holds for both the weakly-supervised and semi-supervised methods. The distribution of pathologies in the set of 200 annotated segmentations is:
> 'Cardiomegaly': 68, 'Enlarged Cardiomediastinum': 109, 'Edema': 45, 'Support Devices': 107, 'Atelectasis': 80, 'Pneumothorax': 8, 'Pleural Effusion': 67, 'Consolidation': 33, 'Lung Lesion': 1
>
> *Bullet 4:*
> We will add a table summarizing the different configurations used by each experiment in the revised manuscript.
>
> *Bullet 5:*
> The inputs in figure 1 represent the segmentation labels in the train set which are used for training the segmentation model. We will update the figure to also include the image itself as an input.
>
> *Bullet 6:*
> We utilized CheXpert segmentation annotations gathered by another piece of work. These annotations will be released soon and will be linked from our codebase.
>
> *Bullet 7:*
> Currently, we are using distillation as a semi-supervised baseline which does not rely on an existing classification model and saliency maps. However, we can certainly combine the approaches and analyze the results.
>
> *Bullet 8:*
> We hypothesize that MoCo-CXR weights do not significantly outperform ImageNet because it is using self-supervised learning instead of fully-supervised learning. We expect that the small performance improvement we see with MoCo-CXR is due to the “a priori” initialization as you mention. In our revised manuscript, we will make sure to note this.
>
> With the following considerations, we hope that the reviewer can revise their rating.

---

### Meta-Review · Area_Chair1 · 2021-03-29

**Recommendation:** Accept (Poster)

**Metareview:**

The motivation for using semi-supervised learning in medical imaging is clear in order to reduce the cost of acquiring dense expert annotations. This paper presents an interesting study on applying recent semi-supervised learning methods to chest x-ray segmentation. There are some conflicting reviews here but in general, the reviewers find the work interesting enough to be presented but with too limited technical novelty to justify an oral presentation.

**Paper Type:**

both

---

### Decision · Program_Chairs · 2021-03-31

Accept